# A Rapid Systematic Review of Factors Influencing COVID-19 Vaccination Uptake in Minority Ethnic Groups in the UK

**DOI:** 10.3390/vaccines9101121

**Published:** 2021-10-01

**Authors:** Atiya Kamal, Ava Hodson, Julia M. Pearce

**Affiliations:** 1Department of Psychology, Birmingham City University, Birmingham B4 7BD, UK; 2Department of War Studies, Strand Campus, King’s College London, London WC2R 2LS, UK; ava.hodson@kcl.ac.uk (A.H.); julia.pearce@kcl.ac.uk (J.M.P.)

**Keywords:** vaccine uptake, minority ethnicity, facilitator, barrier, COVID-19

## Abstract

COVID-19 has disproportionately affected minority ethnic groups in the United Kingdom. To maximise the effectiveness of the vaccination programme, it is important to understand and address disparities in vaccine uptake. The aim of this review was to identify factors influencing COVID-19 vaccination uptake between minority ethnic groups in the UK. A search was undertaken in peer-reviewed databases, polling websites and grey literature from January 2020–May 2021. Studies were included if they reported data on vaccine uptake or the reasons for or against accepting the COVID-19 vaccination for minority ethnic groups in the UK. Twenty-one papers met the inclusion criteria, all of which were rated as either good or moderate quality. Ethnic minority status was associated with higher vaccine hesitancy and lower vaccine uptake compared with White British groups. Barriers included pre-existing mistrust of formal services, lack of information about the vaccine’s safety, misinformation, inaccessible communications, and logistical issues. Facilitators included inclusive communications which address vaccine concerns via trusted communicators and increased visibility of minority ethnic groups in the media. Community engagement to address the concerns and informational needs of minority ethnic groups using trusted and collaborative community and healthcare networks is likely to increase vaccine equity and uptake.

## 1. Introduction

COVID-19 has disproportionately affected minority ethnic groups in the United Kingdom (UK) [1,2]. Ethnicity is a risk factor for adverse outcomes along with other factors such as age, deprivation and comorbidities [1]. However, there is evidence to indicate that some minority ethnic groups have lower intentions to receive COVID-19 vaccinations [3]. This is of concern due to the higher COVID-19 incidence, morbidity and mortality in these groups, and the importance of the vaccination programme as part of a package of measures to manage the pandemic.

Vaccine hesitancy is defined as a delay in acceptance or refusal of vaccination when vaccination services are available [4]. It is characterised by uncertainty and ambivalence and is a legitimate response to fears of safety, concerns about the efficacy of the vaccine, and issues of mistrust towards formal services [5]. 

In England and Wales, people from Asian ethnic groups make up the second largest percentage of the population (7.5%), followed by Black ethnic groups (3.3%), Mixed/Multiple ethnic groups (2.2%) and Other ethnic groups (1.0%) [6]. The UK’s vaccination programme has the potential to exacerbate pre-existing inequalities that the pandemic has exposed and amplified if it does not take into consideration the unequal impact of the pandemic on minority ethnic groups and the factors that enable or hinder vaccination uptake in these groups [7]. To maximise the effectiveness and impact of the vaccination programme, it is important to understand reasons for disparities in uptake which can inform the provision of support for diverse communities, including implications for developing effective public health messaging strategies [5,8].

There is a plethora of vaccination-related evidence across a number of health conditions that predate the COVID-19 pandemic. This evidence has informed early insights such as the likelihood of lower COVID-19 vaccination uptake in minority ethnic healthcare workers based on previous vaccination programmes [5]. Empirical data on COVID-19 vaccination hesitancy and uptake is increasing, with early studies indicating lower vaccine uptake in minority ethnic groups [3]. A synthesis of these studies may offer important insights for each phase of the vaccination programme.

### 1.1. Review Aims

Primary aim: to identify differences in COVID-19 vaccination intention and uptake between minority ethnic groups in the UK.

Secondary aim: to identify barriers and facilitators of COVID-19 vaccination intention and uptake in minority ethnic groups in the UK.

### 1.2. Review Questions

1.What are the differences between minority ethnic groups’ intentions and uptake of COVID-19 vaccination in the UK?2.What are the barriers to COVID-19 vaccination acceptance in minority ethnic groups in the UK?3.What are the facilitators to COVID-19 vaccination acceptance in minority ethnic groups in the UK?4.Are there any differences in barriers and facilitators to COVID-19 vaccination acceptance between minority ethnic groups in the UK?

## 2. Method

A rapid review of the literature was undertaken in accordance with PRISMA criteria for systematic reviews [9]. The protocol for this rapid review was registered on PROSPERO (CRD42021253728) [10].

### 2.1. Search Strategy

The search strategy was applied to Web of Science, Ovid, Scopus and APA PsychINFO for peer reviewed literature, websites of the public polling companies YouGov and Ipsos MORI, and websites detailing public, private, academic and third-sector projects. Grey literature was searched using Google Advanced, and the Public Health England Behavioural Science Research Cell. References and forward citations of relevant articles were also searched. Databases were searched from January 2020 to May 2021.

The following search terms were used: ((vaccin* OR inocul* OR immunis*) AND (COVID-19* OR SARS-CoV-2) AND (uptake OR accept* OR hesitan* OR refus* OR confiden* OR decision* OR concern) AND (ethnic minorit* OR ethnic* OR minorit* OR rac* OR Black* OR African* OR Asian* OR Caribbean OR BME OR BAME OR Roma* OR Eastern European OR migrant OR refugee)).

### 2.2. Inclusion and Exclusion Criteria

Inclusion criteria were:

Population: studies and reports that included people from racial and ethnic minority groups in the UK. Studies and reports that did not include racial and ethnic minority group data from the UK were excluded.

Predictors/Exposures: studies that present data on vaccine uptake, the association between psychological predictors and COVID-19 vaccination, or data on reasons for/against accepting COVID-19 vaccination.

Outcome: COVID-19 vaccination intention or behavior.

Types of studies: original research, quantitative research, qualitative research. Papers were excluded if there was no empirical data, if they reported conference proceedings and/or were not in English.

### 2.3. Data Extraction

For each study or report, details concerning country and region, data collection period, study aims, design, outcome of interest, sampling and recruitment, data collection methods, population characteristics, results, barriers and facilitators of vaccination uptake, and recommendations, were extracted.

### 2.4. Risk of Bias

Risk of bias was assessed using the Mixed Methods Appraisal Tool (MMAT) [11] which evaluates studies on five dimensions based on the study methods. Scoring was adapted in line with a review by Smith, Hodson and Rubin [12] to provide a measure of risk of bias from 1–10. Studies that scored five or under were rated as poor quality, six or seven were moderate quality, and a score of eight or over was rated as good quality. The MMAT was selected as it is a critical appraisal tool designed for systematic reviews that include qualitative, quantitative and mixed-methods studies. While other risk of bias tools are available, the MMAT was selected as it has been validated in several studies and updated in 2018 based on the findings from a literature review of critical appraisal tools, empirical research, and an e-Delphi study with international experts [11].

### 2.5. Procedure

A.K. and J.P. developed the literature search. Screening of titles, abstracts and full-texts, data extraction and risk of bias assessment were carried out by A.H. and A.K. independently screened a sample of titles, abstracts and full-texts, checked a sample of data extracted for accuracy and completeness, and conducted a risk of bias assessment for a sample of studies. Any discrepancies were discussed and resolved through discussion. A.K. conducted the narrative synthesis.

## 3. Results

### 3.1. Search Results

The searches identified 2662 records which were screened, and 29 full-text articles were assessed for eligibility. Of these, 21 were judged to meet the eligibility criteria (see Figure 1).

Seventeen of the 21 studies used quantitative methods, two studies used qualitative methods and two studies used mixed methods. Of the quantitative studies, the majority were cross-sectional surveys (*n* = 11), and the remaining were cohort studies (*n* = 6).

Studies were conducted in all countries within the United Kingdom: England (*n*=3), Northern Ireland (*n* = 1), Scotland (*n* = 2), and Wales (*n* = 1). Within England, two studies were conducted in London (*n* = 2), and one study was conducted in Bradford and North East, North West, Yorkshire and the Humber, East Midlands, West Midlands, South East and South West (see Table 1 for further details).

An overview of population characteristics is presented in Table 1. Classification of ethnic groups varied with studies reporting findings on Black, Asian and Minority Ethnic (BAME) groups (*n* = 4), Black (*n* = 7), Black African (*n* = 7), Black Caribbean (*n* = 7), South Asian (*n* = 3), Indian (*n* = 10), Pakistani (*n* = 8), Bangladeshi (*n* = 8), Pakistani/Bangladeshi (*n* = 1), Asian (*n* = 1), Chinese (*n* = 7), White Irish (*n* = 4), White Other (*n* = 10), Arab (*n* = 3), Mixed (*n* = 11), and Other (*n* = 11) groups.

The age range across all studies was 18-99 years. Topics investigated were vaccine—uptake (*n* = 6), hesitancy (*n* = 10), attitudes (*n* = 4), intention (*n* = 6), and rollout (*n* = 1).

### 3.2. Risk of Bias Analysis

Using the MMAT, the mean average risk of bias score was eight from a maximum of ten (where a higher score means lower risk of bias). Based on the available information, 14 of the studies were rated as good quality, seven as moderate quality, and no studies were rated as poor quality (see Table 1).

### 3.3. Narrative Synthesis

A narrative analysis of the results of vaccination intention and uptake between minority ethnic groups in the UK was conducted. Due to heterogeneity of data, it was not possible to conduct a meta-analysis.

#### 3.3.1. Differences in Minority Ethnic Groups’ Intention and Uptake

Ethnic minority status was associated with vaccine intention and uptake. Thirteen studies reported higher hesitancy and lower intention to get vaccinated in minority ethnic groups compared with White British groups [13,14,15,16,17,18,19,20,21,22,23,24,25] and six studies reported lower vaccine uptake in minority ethnic groups [3,21,26,27,28,29].

Differences between minority ethnic groups were evident with twelve out of 21 studies reporting higher hesitancy and lower uptake in Black groups in comparison with other minority ethnic groups [3,16,19,20,21,22,23,25,26,27,28,29]. Of these twelve studies, six reported the results for subgroups. In three studies, Black African groups had the highest hesitancy and lowest uptake [26,28,29] and three studies reported Black Caribbean groups were most hesitant followed by other Black groups [16,20,27]. Differences between South Asian groups also exist with Pakistani groups reporting the highest hesitancy, followed by Bangladeshi and Indian groups. Higher uptake and lower hesitancy were reported in Indian [16,19,20,25,26,27], Bangladeshi [19,20,27] and Chinese [16,19,25] groups compared with other ethnic minority groups. While hesitancy was lower and uptake was higher for Indian, Bangladeshi and Chinese groups compared with other minority groups, the uptake figure was still lower than in White British groups (see Table 2).

#### 3.3.2. Barriers to COVID-19 Vaccination Acceptance in Minority Ethnic Groups

Barriers to COVID-19 vaccination acceptance were reported across the following domains: psychosocial factors, communications, and practical challenges (see Figure 2).

##### Psychosocial Factors: Trust, Concerns and Beliefs

Heightened vaccine hesitancy was attributed to mistrust including pre-existing lower scientific or medical trust, conspiracy suspicions and attitudes [13,20,21,23,25,30]. Mistrust that results in hesitancy to take the vaccine was reportedly the consequence of negative past experiences that individuals, their family and friends have experienced with formal services [31]. Concerns also included the potential lack of equity for those who choose not to have the vaccine, and that mandating vaccination could create ethnic and racial divides between communities and increase stigma and discrimination [20].

The speed with which vaccines were approved has raised suspicions over whether regulatory standards, meant to protect vulnerable populations, were relaxed for expediency [21]. Under-representation of individuals from ethnic minority backgrounds in vaccine trials also reportedly contributes to hesitancy, along with lack of inclusion of marginalised communities throughout the pandemic [20]. Concerns about the vaccine included how quickly it was produced, that side effects are not known by individuals or those that are producing them, the vaccine has not had time to be fully tested, the vaccine is unsafe, and lack of knowledge about the vaccine [21,24,25,29,32]. As a result, some people wanted to wait three to six months once this information was known, before accepting a vaccine [32].

##### Communications: Source, Content and Access

Misinformation, complex information, conflicting and changing guidance, overwhelming amounts of material, and contradiction of information between different information sources contributed to a lack of trust, confusion, and ultimately vaccine hesitancy [20,30,31,32]. Positively framed vaccine information accessed via mainstream media, Government or National Health Service (NHS) sources fuelled suspicions and contributed to mistrust due to concerns of lack of transparency of risks [20].

Negative vaccine attitudes resulted from various information sources that are alternatives to mainstream media, such as social media [14,31], family which plays a big part in the decision to have a vaccine for some minority ethnic groups [16,20,23], and obtaining information from country of origin [30].

People from minority ethnic backgrounds were more likely than White British groups to have received misinformation encouraging them not to have the vaccine [16,30,31,32]. Some studies cited anecdotes that increased vaccine hesitancy [25], including rumours that ethnic groups are being targeted to test the vaccine or used to harm them [30,32], and concerns about very negative side effects. This was exacerbated by engagement with social media stories which resulted in confusion about whether or not to have the vaccine [32]. Migrants reported specific views ranging from acceptance to misinformation, often originating from social media or word of mouth [30].

Lack of access to information also resulted in communication barriers largely due to low health literacy, poor other language provision, and increased digitalisation of communications. This was particularly an issue for migrant groups due to lack of access to, or knowledge of, technology [20,30].

##### Practical Issues: Logistics

Barriers to vaccination go beyond psychosocial factors and communication. Logistical and practical barriers, such as the location of vaccine centres and having to use public transport, were also reported as barriers to vaccine uptake [18].

#### 3.3.3. Facilitators to COVID-19 Vaccination Acceptance in Minority Ethnic Groups

Facilitators to COVID-19 vaccination acceptance were reported in relation to trust in communication and vaccine outcomes (see Figure 3).

##### Trusted Communication: Family, Social and Professional Influences

More information about the vaccine, including information about the effectiveness, side effects and ingredients, was identified as a facilitator to reduce hesitancy and increase acceptance [19,23]. Respondents from minority ethnic groups said they would trust the view of someone on social media more than they would trust politicians delivering information about the COVID-19 vaccine’s safety and effectiveness [23]. Trusted sources varied and were more likely to be individuals that respondents identified with, such as family, friends, community members and religious leaders [16,20,23,31]. Proactive engagement of healthcare workers from diverse ethnic backgrounds, and increased visibility of less well-represented groups in the media, were also identified as facilitators of vaccine acceptance [20,23,31].

##### Risk Perception: Confidence in Risk Reduction Outcome of Vaccine

Factors influencing risk perception were varied. Personal experience increased perceived risk based on direct experiences (own or others) [18,20]. Specifically, previous infection, knowing people who had been unwell or died from COVID-19, and concerns about infection of their families and loved ones, increased perceived risk [20]. Hearing positive things about the vaccine and addressing concerns of vaccine safety, such as evidence that the vaccine is safe, lowers risk of catching COVID-19 or reduces the risk of being seriously ill, increased confidence in the risk reduction outcome of the vaccine and, consequently, increased vaccination intention [18,22,25,32].

#### 3.3.4. Differences in Barriers and Facilitators between Minority Ethnic Groups

Differences in barriers and facilitators between minority ethnic groups were infrequently reported and pose difficulties in exploring variations beyond broad categories of ‘Black, Asian, White, Other, etc.’. Stated reasons for vaccine hesitancy were often similar across ethnic groups. However, one study noted that, when compared to the White British or Irish group, Black or Black British participants were more likely to state they ‘Don’t trust vaccines’ and the Pakistani or Bangladeshi group cited worries about side-effects [22]. Another study found that the opinion(s) of family and friends were widely trusted and equal with that of a GP in Asian respondents [23].

## 4. Discussion

The results of the review indicate increased hesitancy and lower intention to accept the COVID-19 vaccine in UK based minority ethnic groups compared with the White British population. This is similar to other vaccination programmes and suggests a consistency of concerns beyond COVID-19, for people from minority ethnic groups, such as negative experiences and mistrust towards formal services.

This review highlights a multitude of factors influencing vaccine hesitancy and uptake both across and within minority ethnic groups. Barriers such as pre-existing mistrust of formal services, lack of knowledge and information about the vaccine’s safety, misinformation, complex and changing guidance, inaccessible communications, conflicting information from different information sources, and practical barriers such as the location of vaccine centres, contribute to hesitancy. The results of the narrative synthesis show that Black groups are more likely to cite mistrust of vaccines; this is possibly linked to historical issues of medical malpractice towards Black communities, such as the Tuskegee Syphilis study [5]. Pakistani or Bangladeshi groups have concerns about possible side effects that could be explained by inaccessible communications for some of these groups [8]. Accordingly, facilitators include addressing vaccine concerns via trusted communicators and increased visibility of minority ethnic groups in the media and could be broadened further by delivery of healthcare messages in a range of languages [33].

Since the start of the vaccination programme, work is underway with communities to address increased vaccine hesitancy and lower vaccine uptake in minority ethnic communities. Strategies include a nationally funded Community Champions programme [34,35] which supports local public health teams to work with communities to address local barriers and facilitators, tailored communications and local monitoring and insights [36].Vaccine uptake has increased across all minority ethnic groups since the start of the vaccination programme. In the 50+ age group, vaccine uptake has increased from 73.1% to 81.7% for Pakistani groups, from 83.1% up to 89.5% for Bangladeshi groups, from 61.6% to 67% for Black Caribbean groups, and from 64.9% to 73.5% for Black African groups [37]. A similar pattern is evident in younger age groups [37]. The gap in vaccine intention and uptake between White British and minority ethnic groups has closed but minority ethnic groups still have a lower proportion of vaccine acceptance and uptake [37], which indicates ongoing work with communities is required at each stage of the vaccination programme. A lower proportion of minority ethnic groups receiving a COVID-19 vaccine may not be an issue that is unique to the UK, as Black and Hispanic groups in the United States of America also have lower vaccine uptake. This has been attributed to similar reasons relating to mistrust, practical barriers and systemic challenges [38]. Data on ethnicity is not routinely collected or reported in some countries, which limits further cross-national comparisons.

Underpinning the issues identified in this review is the notion that minority ethnic groups should not be treated as one homogenous group and the need to adopt a person-centred approach to support specific communities by addressing their unique needs and concerns about the COVID-19 vaccine [5,16].

### 4.1. Recommendations

The following recommendations are based on the studies included in this review:

#### 4.1.1. Community Engagement

Vaccine hesitancy is driven, in part, by anxiety fuelled by misinformation or lack of information. There is a need for clear, honest and responsive information that is sensitively framed and nonjudgemental to overcome this [17]. Many respondents from minority ethnic groups that do not want the vaccine would reconsider their decision if they had more information [19]. It is therefore essential to engage with minority ethnic communities to understand what information is required, when, where, and in what format; to coproduce vaccine communication campaigns in a range of media and languages and establish a two-way dialogue directly with people to respond to questions or concerns, and to tackle misinformation [17,19,20,22,30].

It is important to harness connections that exist within communities to counter misinformation. Promoting vaccination via trusted networks such as religious leaders, and people within community support roles such as teachers, nursery workers and advice workers, can ensure the spread of correct information [8,17,32]. In addition to community engagement, proactive engagement with healthcare workers, particularly those from diverse backgrounds, can increase visibility of less well represented groups and build trust [20,31].

Engagement opportunities for those that are undecided about the vaccine is essential, and collaborative working between community led-events and medical professionals can provide access to educational resources such as information about the UK’s health and social care system, clinical trials and the regulation process. This is particularly important in addressing the legacy of historic unethical research and issues faced by Black people in clinical trials [31].

#### 4.1.2. Inclusive Communications

Public health messages should highlight the prosocial benefits of a vaccine and draw on collective identities that are sensitively attuned to different communities [17], but should avoid negative assumptions and stereotyping associated with ethnicity as this may marginalise and stigmatise communities [17,39]. Inclusion of people from different racial and ethnic minority backgrounds in posters, videos and communications will increase opportunities for minority ethnic groups to identity with health messages in a more relatable way; this includes distributing information that has been translated [17,20,30,31].

#### 4.1.3. Practical Application and Service Delivery

The views of minority ethnic communities, once obtained via community engagement efforts, should be integrated into NHS and Public Health England (PHE) teams [19]. Practical barriers, such as location or cost implications, should be addressed to ensure equity in accessibility and opportunity to have the vaccine [20]. Innovations in service delivery, such as translated health advice using online platforms that translate resources and provide language specific advice, and use of multiple communication channels such as text, email, letter and posters in local community hubs, may reduce accessibility issues [30].

Local targeted responses to misinformation can be achieved quickly if processes are in place for systematic monitoring of the circulation of misinformation on social media [17].

#### 4.1.4. Resources

Innovations in service delivery, and utilising the expertise of local community groups to codesign delivery approaches and communications, requires time, planning and financial support. Resourcing support is required for the availability of interpreters and culturally appropriate translations of vaccine advice, alongside community ambassadors, peer support, Community Champions in vaccine centres, and development and implementation of local communication campaigns and events [30,36]. Healthcare professionals should also be provided with resources and time to address the concerns of individual patients about the Covid-19 vaccine [23].

#### 4.1.5. Further Research

Ongoing engagement with minority ethnic groups, including subgroups such as young people, is required to understand and address the barriers to vaccine uptake. Further insight into the barriers and facilitators experienced by different groups will support localised, targeted responses to vaccine concerns. Initiatives to improve uptake in Black ethnic groups within the UK should be an urgent priority, including qualitative explorations of the reasons for vaccine hesitancy and the approaches to overcoming them [22].

### 4.2. Strengths and Limitations of the Review

This review included 21 studies reporting data on UK minority ethnic groups’ COVID-19 vaccine intention and uptake. The review includes data that reflects the earlier stages of the vaccination programme, during which older adults and frontline workers were prioritised. It was not possible to synthesise findings based on age, as several papers included in the review did not report such data. In papers where age was reported, this often ranged from 18 to 99 years with a mean age of approximately 46 years. It is important for future research to stratify results based on age, as younger groups may have different reasons for hesitancy in comparison to older groups, which was not possible to establish in this review. As the vaccination programme has progressed, it is likely that some of the respondents who were initially hesitant due to concerns about side effects may hold more positive attitudes and may now believe that the vaccine is safe. For example, recent data shows higher vaccine uptake compared with lower intentions reported in 2020 before the vaccination programme was available [37,40]. However, the implications for policy and practice remain unchanged and can support minority ethnic groups at each stage of the vaccination programme, including offering booster jabs and supporting younger populations.

Lower COVID-19 vaccine intentions and uptake was defined based on the proportion of responses in comparison to White British groups. This is a limitation of the review as the gap between vaccine intention and uptake varied across studies and was not weighted or standardized. As a result, one study may have reported a higher proportion of hesitancy compared with another study, but both were considered equally in terms of vaccine hesitancy.

It was not possible to identify regional differences due to small sample sizes; only two included studies reported regional data in England for some regions [7,9]. Small numbers of respondents in minority ethnic groups did not allow for detailed analysis in some of the studies, which means broad categories of ‘BAME’ were used [22,23]. In addition to this, the categories for ethnicity self-identification differed across studies and posed difficulties in integrating and synthesising the study findings at a more detailed level. For example, data for some groups was reported as the broad category ‘BAME’, other studies reported findings for Black or South Asian groups with no distinction between within-group differences such as Black Caribbean or Black African or Pakistani or Indian, whereas other studies did make the distinction between these groups. Standardisation of specific ethnic group categories may provide more detailed insights and facilitate group comparisons in a more meaningful way that can support the design and delivery of vaccination programmes.

## 5. Conclusions

It is important to build on the initial success of the UK vaccination programme to achieve sufficient protection at a population level. Although the majority of the eligible UK population has received a COVID-19 vaccine since the vaccine programme was launched in December 2020, including 89.1% of the population who have received a first dose and 81% who have received both doses, [41] vaccination rates remain lower in minority ethnic groups, including younger age groups. It is essential that the views of minority ethnic communities are understood by the NHS and local and central government partners, and are implemented in vaccine service design and delivery. If the concerns of communities are not considered and addressed, there is a risk that the vaccination rollout could exacerbate inequalities by providing inaccessible or inappropriate support. This review identified vaccine hesitancy as multifaceted, but support embedded into specific communities can address the concerns and informational needs of minority ethnicity groups. Using trusted and collaborative healthcare and community networks is likely to increase both vaccine equity and accessibility to vaccinations.

## Figures and Tables

**Figure 1 vaccines-09-01121-f001:**
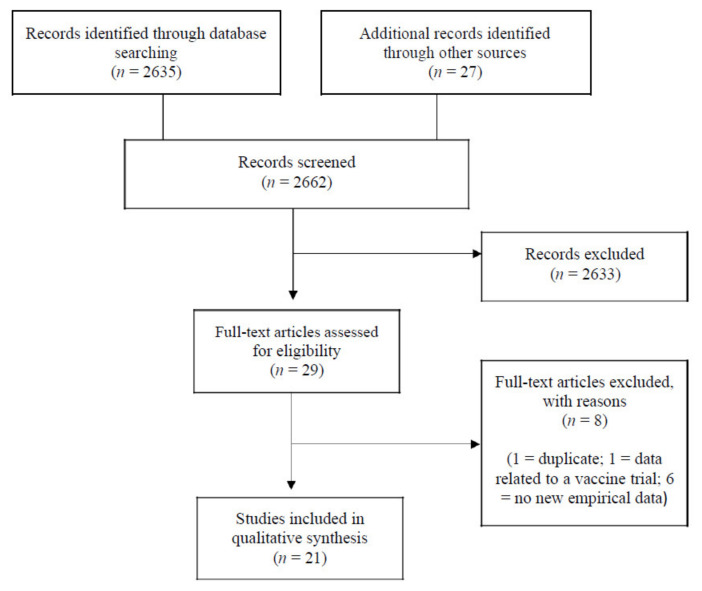
PRISMA Flowchart depicting the selection of studies for the rapid systematic review.

**Figure 2 vaccines-09-01121-f002:**
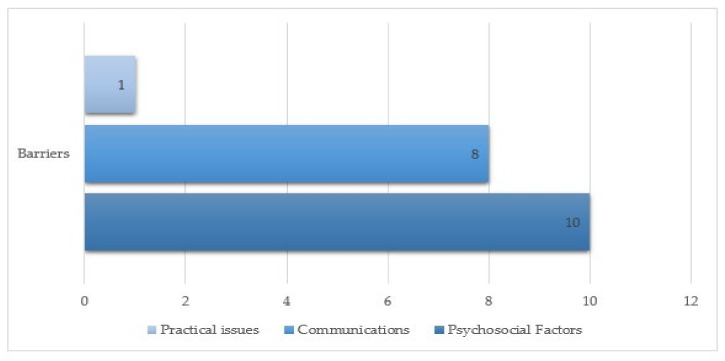
Distribution of barriers to COVID-19 vaccination.

**Figure 3 vaccines-09-01121-f003:**
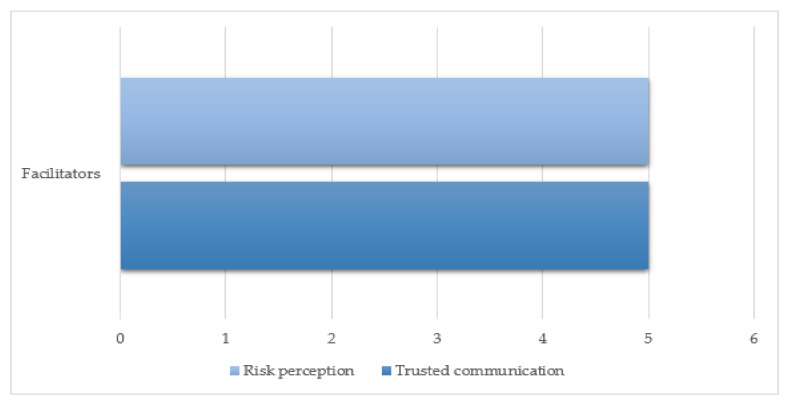
Distribution of facilitators of COVID-19 vaccination.

**Table 1 vaccines-09-01121-t001:** Study Characteristics.

First Author, Date	Country, Region	Study Design	Data Collection Period	Population Characteristics	Topics Investigated	Top-Line Findings	Risk of Bias Score
Allington, 2021a	UK	Quantitative	November— December 2020	*n* = 4343 Gender (F) = 51% Age (mean) = 46.23 Ethnic group (White = 91%, Other = 7%) * weighted averages reported	Vaccine hesitancy	Ethnic minority status is associated with vaccine hesitancy, conspiracy suspicions, use of social media for information about coronavirus, negative vaccine attitudes and age. Very weak correlation between membership of an other-than-white ethnic group and trust in scientists, and slightly stronger negative correlation with trust in medical professionals.	9 (Good)
Allington, 2021b	UK	Quantitative	Study 3 and Study 4: June 2020	Study 3: *n* = 1663 Age (mean) = 50.0 Gender (F) = 55.1% Ethnicity (White = 94.3%, Other = 5.7%) Study 4: *n* = 2237 Age (mean) 44.5 Gender (F) 49.7% Ethnicity (White = 89.7%, Other = 10.3%)	Vaccine intention	Study 3: voting history and frequency of legacy media consumption associated with SARS-CoV-2 vaccine intentions. Vaccine intentions negatively associated with other than white ethnicity. Study 4: voting history and informational reliance on legacy and social media associated with SARS-CoV-2 vaccine intentions. Vaccine intentions negatively associated with other than white ethnicity.	9 (Good)
Bell, 2020	England	Mixed-methods	Survey: April–May 2020 Interviews: April–May 2020	Survey: *n* = 1252 Gender: Female (95.0%; *n* = 1190) White British 1082 (86.4) White Irish 20 (1.6) White Other White background 76 (6.1) Black or Black British—African 3 (0.2) Black or Black British—Caribbean 1 (0.1) Mixed: White and Black Caribbean 7 (0.6) Mixed: White and Black African 1 (0.1) Mixed: White and Asian 9 (0.7) Mixed: Other mixed background 7 (0.6) Asian or Asian British—Indian 15 (1.2) Asian or Asian British—Pakistani 10 (0.8) Asian or Asian British—Bangladeshi 3 (0.2) Asian or Asian British—Other Asian background 3 (0.2) Chinese 2 (0.2) Other ethnic group 7 (0.6) Do not wish to say 6 (0.5) Interview: *n* = 19 Age (mean) = 33.4 White British (11) Other White background (3) Mixed White and Black Caribbean (2) Mixed White and Asian (1) Asian or Asian British: Pakistani (1) Chinese (1)	Vaccine intention	Black, Asian, Chinese, Mixed or Other ethnicity participants were 2.7 times more likely to reject a COVID-19 vaccine for themselves and for their child than White British, White Irish and White Other participants.	9 (Good)
British Red Cross, 2021	UK	Quantitative	February 2021	UK nationally representative sample (*n* = 2005) Boost of ethnic minorities (excluding White ethnic minorities) sample (*n* = 1000) Boost of people from Black ethnic groups (*n* = 503) Boost of people from South Asian ethnic groups (*n* = 508)	Vaccine hesitancy	Minority ethnic respondents are more likely to get vaccine information from friends and family, trust family more than other sources of vaccine information other than health professionals, discuss their decision about having the vaccine with extended family, and are more likely to have received information encouraging them not to have the vaccine.	9 (Good)
Caribbean and African Health Network, 2021	UK	Quantitative (survey with open-ended question responses)	January 2021	*n* = 220	Vaccine attitudes, vaccine intention	Vaccine knowledge and intention improved after an educational session.	7 (Moderate)
Freeman, 2020	UK, North East 192 (3.8%) North West 567 (11.1%) Yorkshire and the Humber 414 (8.1%) East Midlands 357 (7.0%) West Midlands 470 (9.2%) East 405 (7.9%) London 723 (14.1%) South East 731 (14.3%) South West 427 (8.3%) Wales 257 (5.0%) Scotland 465 (9.1%) Northern Ireland 106 (2.1%)	Quantitative	September–October 2020	*n* = 5114 Age: 18−24 571 (11.2%); 25−34 898 (17.6%); 35−44 883 (17.3%); 45−54 929 (18.2%); 55−64 761 (14.9%); 65−99 1072 (21.0%). Gender: male 2574; female 2515; nonbinary 20; prefer not say 5. White: English/Welsh/Scottish/Northern Irish/British 4056 (79.3%); Irish 57 (1.1%); Gypsy or Irish Traveller 8 (0.2%); any other White background 204 (4.0%). Mixed: White and Black Caribbean 43 (0.8%); White and Black African 17 (0.3%); White and Asian 34 (0.7%); any other Mixed/Multiple ethnic background 27 (0.5%). Asian/Asian British: Indian 146 (2.9%); Pakistani 105 (2.1%); Bangladeshi 50 (1.0%); Chinese 49 (1.0%); any other Asian background 51 (1.0%). Black/African/Caribbean/Black British: African 128 (2.5%); Caribbean 71 (1.4%); any other Black/African/Caribbean background 16 (0.3%). Other ethnic group: Arab 13 (0.3%); Any other ethnic group 13 (0.3%).	Vaccine hesitancy	Vaccine hesitancy is associated with Black or mixed ethnicity	9 (Good)
Healthwatch, 2021	Not available	Quantitative	January 2021	*n* = 2431	Vaccine hesitancy	Vaccine hesitancy exists among Black, Asian, and minority ethnic groups	7 (Moderate)
Healthwatch Camden, 2021	Camden, London	Quantitative	February 2021	*n* = 223 (97% BAME) Arab (~1%) Black African & Asian (~1%) Black African & White (~1%) Asian/Asian British: Indian (~2%) Black/Black British: Caribbean (~2%) Asian & White (~2%) Other ethnic background (~2%) Any other White background (~3%) White: British/N. Irish/Scottish/Welsh (~4%) Any other Black/Black British background (~5%) I’d prefer not to say (~8%) Asian/Asian British: Chinese (~9%) Black/Black British: African (~12%) Black/Black British: Somali (~20%) Asian/Asian British: Bangladeshi (~27%)	Vaccine attitudes, vaccine hesitancy	Vaccine hesitancy higher among Black, Asian, and minority ethnic groups compared to national data.	6 (Moderate)
Knights, 2021	England	Qualitative	June–November 2020	*n* = 81 Phase 1: *n* = 64 clinicians and administrative staff Age (mean) = 45 Gender (F) = 54 (84.4%) White British 32 (50.0%) White Irish 3 (4.7%) African 4 (6.2%) Caribbean 1 (1.6%) Indian 11 (17.2%) Pakistani 3 (4.7%) Other Asian background 2 (3.1%) Other Mixed background 3 (4.7%) Other White background 5 (7.8%) Phase 2: *n* = 17 migrants Age (mean) ** = 37.9 Gender (F) = 11 (64.7%) WHO Region of Origin (%) = African (Mauritius, Nigeria, Zimbabwe, other) 4 (23.5%), The Americas (Venezuela) 1 (5.9%) Eastern Mediterranean (Afghanistan, Egypt, Iraq, Pakistan, Palestine) 3 (17.6%) South East Asian (Sri Lanka) 4 (23.5%)	Vaccine roll-out	Pre-existing distrust of vaccinations and the NHS alongside low health literacy and widespread misinformation are likely to negatively affect uptake of a potential COVID-19 vaccine in some migrants. Migrants reported views ranging from acceptance to misinformation, often originating from social media or word of mouth.	10 (Good)
Lockyer, 2021	Bradford, UK	Qualitative	September–October 2020	*n* = 20 Age (range 20–85) most aged 25–54 Asian or Asian British (Pakistani, Indian and Bangladeshi) (10) White British (6) White Other (Eastern European, Gypsy or Irish Traveller) (4)	Vaccine hesitancy	Mixed findings: nine would accept a COVID-19 vaccine (with caveats around safety), five felt mixed, and six were not willing to have it. Results were not reported by ethnicity.	10 (Good)
Nguyen, 2021	UK	Quantitative	March 2020–February 2021	*n* = 4,427,024 White (*n* = 1,204,721) Black (*n* = 9615) South Asian (*n* = 17,628) Middle East/East Asian (*n* = 7689) More than one/other (*n* = 14,641) Sex (F) and ethnicity: White *n* = 71,1693 (59.1) Black *n* = 5642 (58.7) South Asian *n* = 9883 (56.1) Middle East/East Asian *n* = 4438 (57.7) More than one/other *n* = 9016 (61.6)	Vaccine hesitancy, vaccine uptake	95% of all participants willing to accept vaccine. Racial/ethnic minorities were more likely to report being unsure or unwilling to undergo vaccination. Black frontline healthcare workers had lower vaccine uptake than their White counterparts.	8 (Good)
Office for National Statistics, 2021a	England	Quantitative	December 2020–March 2021	Age group (vaccination rate %): 70–74 (88.6%), 75–79 (91.5%), 80–84 (92.6%), 85–89 (91.0%), 90–94 (88.0%), 95–99 (83.0%), 100+ (71.1%) Sex (vaccination rate %): Female (90.4%), Male (90.0%) Ethnic group (vaccination rate %): White British (91.3%) Indian (86.2%) White other (81.6%) Mixed (80.4%) Other (77.6%) Chinese (76.7%) Pakistani (74.0%) Bangladeshi (72.7%) Black Caribbean (68.7%) Black African (58.8%)	Vaccine uptake	The percentage vaccinated was lower among all ethnic minority groups compared with the White British population.	7 (Moderate)
Office for National Statistics, 2021b	England	Quantitative	December 2020– April 2021	Age group (vaccination rate %): 50–59 (89.0%), 60–69 (92.9%), 70–79 (93.2%), 80–89 (96.3%), 90+ (94.9%) Sex (vaccination rate %): Female (92.4%), Male (91.6%) Ethnic group (vaccination rate %): White British (93.7%) Indian (90.9%) Bangladesh (86.9%) Chinese (83.7%) Mixed (81.2%) Other (80.8%) White other (80.8%) Pakistani (78.4%) Black African (71.2%) Black Caribbean (66.8%)	Vaccine uptake	Among people aged 50 years and over, vaccination rates for the first dose of a COVID-19 vaccine were lower for all ethnic minority groups when compared with White British group.	7 (Moderate)
Office for National Statistics, 2021c	UK	Quantitative	March–April 2021	All adults (16,360) White (15,240) Asian or Asian British (550) Black or Black British (220) Mixed (210) Other ethnic group (120)	Vaccine uptake, vaccine hesitancy	The percentage vaccinated was lower among all ethnic minority groups compared with the White population.	7 (Moderate)
Robertson, 2021	UK	Quantitative	November–December 2020	*n* = 12,035 (12,035 participants completed the Covid-19 wave 6 survey online and the weighted sample is 9981) Gender: Male 4666 (46.8%) Female 5290 (53.0%) Prefer not to say 25 (0.3%) Age: 16–24 920 (9.2%), 25–34 1382(13.8%), 35–44 1543 (15.5%), 45–54 1784 (17.9%), 55–64 1938 (19.4%), 65–74 1532 (15.3%), 75+ 882 (8.8%) White British or Irish 8713 (87.3%) Other White background 269 (2.7%) Mixed 168 (1.7%) Asian or Asian British—Indian 176 (1.8%) Asian or Asian British—Pakistani/Bangladeshi 198 (2.0%) Asian or Asian British—any other group 106 (1.1%) Black or Black British 190 (1.9%) Other Ethnic Group 59 (0.6%) Missing 102 (1.0%); Born in UK: Born in UK 8991 (90.1%) Not Born in UK 824 (8.3%) Missing 166 (1.7%) UK Country: England 8424 (84.4%) Wales 507 (5.1%) Scotland 775 (7.8%) Northern Ireland 275 (2.8%)	Vaccine hesitancy	Higher vaccine hesitancy was seen in most minority ethnic groups compared to the White British or Irish group.	10 (Good)
Royal Society for Public Health, 2020	UK	Quantitative	December 2020	*n* = 2076	Vaccine attitudes, vaccine intention	Willingness to be vaccinated was lower among people from Black, Asian or Minority Ethnic backgrounds. Respondents from minority backgrounds were more receptive to changing their minds about getting the vaccine if given more information compared with White respondents.	8 (Good)
The OpenSAFELY Collaborative, 2021a	UK	Quantitative	December 2020–January 2021	*n* = 1,160,062 (F = 669,278) White = 788,806 Unknown = 325,637 South Asian = 26,936 Black = 10,329 Other = 5539 Mixed = 2805 Age 80+ 476,375; 70–79 years 74,108; health or social care workers under 70 years 378,921; care home residents aged 65 + 32, 174.	Vaccine uptake	The proportion vaccinated was highest in White groups and lower in all ethnic minority groups.	8 (Good)
The OpenSAFELY Collaborative, 2021b	UK	Quantitative	December 2020–March 2021	*n* = 2,558,906 F = 1,400,532; M = 1,021,944 White 1,515,535 Unknown 923,363. South Asian 65,975 Black 34,517 Other 10,934 Mixed 8554	Vaccine uptake	The proportion vaccinated was highest in White groups and lower in all ethnic minority groups.	8 (Good)
Williams, 2021	Scotland	Quantitative	Time 1: May–June 2020 Time 2: August 2020	Time 1: Age 18–49 (1847, 53.8%); 50+ (1578, 45.9%) Female (2219, 79.1%) Male (666, 19.4%) White (3308, 96.3%) Black, Asian and Minority Ethnic group (101, 2.9%) Time 2: *n* = 2016 (59% follow-up rate) Age 18–49 (947, 48.3%); 50+ (1034, 51.5%) Female (1632, 82.1%) Male (355, 17.9%) White (1949, 96.7%) Black, Asian and Minority Ethnic group (52, 2.6%)	Vaccine intention	Participants of White ethnicity had higher levels of intention than those from Black, Asian and Minority Ethnic groups.	10 (Good)
Woolf, 2021	UK	Mixed-methods	December 2020–March 2021	Healthcare workers. Cohort Study: *n* = 11,584 Age, median (IQR): 45 (34–54) Sex (%): Female 2797 (24.2%) White: 6907 (60.8%); Irish 209 (1.8%); Other/Gypsy Irish Traveller 878 (7.7%). Asian: Indian 1187 (10.4%); Pakistani 315 (2.8%); Bangladeshi 69 (0.6%); Chinese 253 (2.2%); Other 365 (3.2%). Black: African 349 (3.1%); Caribbean 102 (0.9%); Other 20 (0.2%). Mixed: White & Black African 66 (0.6%); White & Black Caribbean 84 (0.7%); White & Asian 179 (1.6%); Other 142 (1.3%). Other: Arab 122 (1.1%); Other 123 (1.1%). Qualitative Study: *n*= 99; 41 interviews (*n* = 24) and focus groups (*n* = 17), and 58 from the longitudinal cohort study (free text comments provided about vaccinations). 27 (66%) were women Ethnicity of qualitative participants— Asian 13 (32%) Black 12 (29%) White 10 (24%) 24 were born in the UK (59%) Ethnicity of the 58 cohort participants: White 42 (72%) Asian 8 (14%) Black 4 (7%) 48 participants (83%) were women	Vaccine hesitancy	Healthcare workers from Black Caribbean, Black African and White Other ethnic groups were significantly more likely to be vaccine hesitant compared to White British healthcare workers.	10 (Good)
YouGov, 2021a	UK Region: London = 400 Rest of South = 200 Midlands/Wales = 210 North = 160 Scotland = 30	Quantitative	February 2021	Male 482; Female 518 Age: 18–24 = 155; 25–49 = 579; 50–64 = 184; 65+ = 82. Black = 241 Indian = 201 Pakistani = 149 Other ethnicity = 118 Other Asian = 110 Mixed = 84 Bangladeshi = 56 Chinese = 40 Country of birth—UK = 370; Outside of the UK = 630.	Vaccine attitudes, Vaccine intention	The majority of respondents had positive intentions of getting a vaccine and a more favourable attitude towards the Pfizer-BioNtech coronavirus vaccine compared with the AstraZeneca and Moderna vaccine.	6 (Moderate)

*Demographic weights were calculated post-collection on the basis of education and geographical region and of gender interlocked with age, NRS social grade and working status. The sample was designed and weighted for demographic representativeness and is treated as equivalent to a random sample. ** Age data missing for two participants.

**Table 2 vaccines-09-01121-t002:** Qualitative barriers, facilitators, differences and recommendations.

First Author, Date	Facilitators	Barriers	Differences between MEGs	Recommendations
Allington, 2021a	-	Conspiracy suspicions Vaccine attitudes Lower scientific and medical trust	-	Further data required to understand reasons for different vaccine attitudes and conspiracy suspicions.
Allington, 2021b	-	Reduced use of legacy media Social media inadequate replacement to legacy media	-	-
Bell, 2020	-	-	-	-
British Red Cross, 2021	Key communications with trusted sources such as healthcare professional and scientists Family conversations	Concerns about side effects Misinformation	Black Caribbean, Black African or Pakistani groups are most likely to reject the vaccine. Indian and Chinese communities are just as likely to have already had/planning to have the vaccine as the UK average.	Individuals from minority ethnic backgrounds should not be approached as one homogenous group. Adopt a person-centred approach to communications.
Caribbean and African Health Network, 2021	Key communications with individuals they identify with and influential others such as Black medical professionals and religious leaders	Perceived health inequalities Misinformation Lower scientific and medical trust	-	Community-led online events with medical professionals. Health messaging and education should target the Black community, with integration of faith and community leaders. Build trust in medical professionals (for example after historic medical mistreatment of Black individuals) and provide education on the UK’s health and social care system (e.g., clinical trials).
Freeman, 2020	Beliefs about collective importance of a COVID-19 vaccine	Mistrust	-	Enhancing and improving health messaging to emphasise prosocial benefits of the vaccine; attuned to different kinds of collective identities; transparent about safety and efficacy.
Healthwatch, 2021	Wait until others have the vaccine before getting it themselves	Logistical issues such as location of vaccination sites and using public transport Mistrust of vaccine programme/rollout	-	Find out what differences exist in attitudes and barriers for different parts of the Black and Asian communities.
Healthwatch Camden, 2021	More information	-	People from Black or Black British (mainly Somali, African, Caribbean, or other Black/Black British) background were more hesitant to get the vaccine. People who identified as Asian or Asian British (mainly Bangladeshi, Chinese, and Indian) had less vaccine hesitancy than the ethnic minority group average.	Integrate views of Black, Asian and Minority Ethnic groups into NHS practice and care. Conduct further research into the informational needs of minority ethnic communities including what information is needed, when, where and in what format. Coproduce communication campaigns.
Knights, 2021	-	Pre-existing distrust of vaccinations and the NHS Concerns about vaccine safety Migrant communities have not been included in trials Low health literacy Misinformation and contradiction of information between different information sources Seeking information from country of origin Digitalisation resulted in lack of access to knowledge and communication barriers Beliefs COVID-19 is a Western disease Fear of discrimination or being used as ‘guinea pigs’ Reliance on home remedies	-	Innovations in service delivery such as translated health advice using text templates and YouTube. Practices should seek to ensure they can identify migrants, that they understand their needs through proactive engagement, and that they are providing language-specific advice about COVID-19 and changes in service provision in the pandemic through multiple modalities (e.g., text, email, letter and posters in local community hubs). Use of patient participation groups and other local community groups to codesign delivery approaches. Ensure availability of interpreters and translated culturally-appropriate vaccine advice, alongside integration of migrant ambassadors into vaccine centres, and information-sharing campaigns.
Lockyer, 2021	Will wait 3–6 months to see what the effects of the vaccine are on others.	Concerns about speed of vaccine development and unknown side effects Social media stories about misinformation and severe side effects which results in confusion and not refusal Rumours that ethnic groups are being targeted to test the vaccine or used to harm them	-	Hesitancy is rooted in anxiety fuelled by misinformation which needs to be mediated by clear, honest and responsive information that is sensitively framed and nonjudgemental. Provide health, social and community workers with an up-to-date summary of locally circulating misinformation with resources to help them counter concerns and provide informed reassurance.
Nguyen, 2021	-	Mistrust of the medical system Lack of diverse representation in clinical trials The speed with which vaccines were approved has raised suspicions over whether regulatory standards meant to protect vulnerable populations were relaxed for expediency	Black groups more hesitant or unsure followed by Middle East/East Asian, South Asian, and Other groups. Vaccine uptake highest in South Asian group and lower in Black healthcare workers.	Need to address long-standing systemic disparities to achieve the health equity required for population-scale immunity.
Office for National Statistics, 2021a	-	-	The lowest vaccination rates were among Black African, Black Caribbean, Bangladeshi and Pakistani groups. The vaccination rate among people from an Indian background was lower than that of the White British group but was high overall.	-
Office for National Statistics, 2021b	-	-	Vaccination rates were lowest for Black Caribbean, Black African and Pakistani groups. Although lower than the White British group, vaccination rates among Indian and Bangladeshi groups remained high.	-
Office for National Statistics, 2021c	-	-	Around 1 in 3 Black or Black British adults reported vaccine hesitancy, the highest compared with all ethnic groups.	-
Robertson, 2021	If the vaccine was demonstrated to be safe and reduced their risk	Lack of trust in vaccines Concerns about side effects	Black or Black British were the ethnic group with the highest rate of vaccine hesitancy followed by Pakistani/Bangladeshi groups and those of Mixed ethnicity. Black participants were more likely to state they do not trust vaccines and Pakistani/Bangladeshi groups cited worries about side effects.	Include subgroups that are hesitant in the planning and development of engagement programmes. Initiatives to improve uptake in Black ethnic groups within the UK should be an urgent priority; for example, by working in close partnership with communities and making use of community champions.
Royal Society for Public Health, 2020	Information about the vaccine from GP or another health professional Information about the effectiveness, side effects and ingredients of the vaccine Social media influences Family and friends influence	Concerns about side effects Vaccine not tested in diverse ethnic groups Mistrust	Lowest willingness to be vaccinated in Asian respondents. The sample sizes were too small to draw any substantive conclusions, but the levels of confidence in the vaccine appear to be broadly similar across minority groups.	Support people to have productive conversations with their peers and relatives about common concerns surrounding the vaccine. Healthcare professionals need to be equipped with both the resources and the time to address the concerns of individual patients about the Covid-19 vaccine. Further research to understand why vaccine confidence is lower in minority ethnic groups.
The OpenSAFELY Collaborative, 2021a	-	-	The proportion vaccinated was lowest in Black people followed by mixed, other and South Asian ethnicities. People from Black African groups had the lowest vaccination rates followed by Other Black and Mixed Black, Caribbean, Pakistani and Bangladeshi groups.	The reasons underpinning variation in vaccination coverage are not yet understood. Further research is needed to understand and address the disparity between ethnic groups.
The OpenSAFELY Collaborative, 2021b	-	-	Vaccination coverage was lowest in all Black groups with the lowest uptake in Black African followed by Mixed White and Black African, any other Black background, Black Caribbean, and Mixed White and Caribbean groups. The lowest uptake in Asian groups was Pakistani followed by Chinese, Any other ethnic group, and Bangladeshi groups.	-
Williams, 2021	-	-	-	A better understanding of the barriers to vaccination in subpopulations and diverse communities is required to collectively be better prepared to deliver appropriate evidence-based culturally and community-appropriate messaging aimed at maximising COVID-19 vaccine uptake.
Woolf, 2021	Vaccine confidence among family, friends and community members Increased risk perception based on previous infection, knowing people who had been unwell or passed away from COVID-19, and concerns about infection of their families and loved ones Increased visibility of less well represented groups in the media More proactive involvement and engagement of healthcare workers from diverse ethnic backgrounds	Complex information, conflicting and changing guidance, overwhelming amounts of material, and poor provision of information in other languages Mistrust of mainstream media Conspiracy beliefs Underrepresentation of individuals from ethnic minority backgrounds in vaccine trials Lack of inclusion of marginalised communities throughout the pandemic	Black Caribbean healthcare workers were most hesitant followed by Mixed White and Black Caribbean, Black African, Chinese, Pakistani, and White Other groups compared with Indian and Bangladeshi healthcare workers who had lower levels of hesitancy.	Develop inclusive communication through a range of media and languages, and engage directly with people to respond to questions or concerns, and tackle misinformation. Use language which avoids assumptions or stereotyping associated with ethnicity. Equity in accessibility and opportunity to have the vaccine is paramount for improving delivery.
YouGov, 2021a	Hearing positive things about the vaccine	Don’t know enough about the vaccine Concerns about safety Lack of trust in science underpinning vaccine Hearing negative stories	Chinese groups had the strongest intention to get vaccinated followed by Indian and Mixed groups, and people in the Black group had the lowest intention to get vaccinated.	-

- Indicates none reported.

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
