# Peer review of "A Rapid Systematic Review of Factors Influencing COVID-19 Vaccination Uptake in Minority Ethnic Groups in the UK"

_vaccines, 2021, doi:10.3390/vaccines9101121_

Round 1

Reviewer 1 Report

This paper provides a clear and comprehensive systematic review of factors influencing COVID-19 vaccination uptake in minority ethnic groups in the UK. The study was well-designed and meaningful.

  1. However, I think this paper need more information to illustrate the importance of studying the influencing factors of ethnic minority vaccination in the background part. Why was the study population ethnic minorities? What percentage of the population are ethnic minorities in UK?Are vaccination rates low among ethnic minorities?
  2. Table 1 can be revised to look better, especially the columns of Population characteristics and Top-line findings.
  3. In the 3.3.2. Barriers to COVID-19 vaccination acceptance in minority ethnic groups part, Could you add some intuitive figures,such as column charts to show the proportional distribution of the barriers to COVID-19 vaccination acceptance?

Author Response

Many thanks for the helpful feedback. We have addressed each of the points raised by reviewer one accordingly:

Point 1. I think this paper need more information to illustrate the importance of studying the influencing factors of ethnic minority vaccination in the background part. Why was the study population ethnic minorities? What percentage of the population are ethnic minorities in UK? Are vaccination rates low among ethnic minorities?

Response: Additional information has been added to the background section which presents ethnicity as a risk factor for adverse outcomes and strengthens the rationale for studying the factors that influence vaccination uptake in ethnic minority groups. Population level statistics have also been included along with evidence that vaccination rates are lower among ethnic minority groups.

Point 2.  Table 1 can be revised to look better, especially the columns of Population characteristics and Top-line findings.

Response: Table 1 has been reformatted to landscape (which is how it was originally intended). The population characteristics and top-line findings have been edited to make the information more succinct and accessible to readers.

Point 3. In the 3.3.2. Barriers to COVID-19 vaccination acceptance in minority ethnic groups part, Could you add some intuitive figures,such as column charts to show the proportional distribution of the barriers to COVID-19 vaccination acceptance?

Response: Figures 2 and 3 have been added which show the proportional distribution of barriers and facilitators to COVID-19 vaccination acceptance.

Reviewer 2 Report

I think this Systematic Review is relevant for the  Covid-19 literature in the UK. The study takes into account the fact the the vaccination uptake had been lower for minority ethnic groups and suggests potential solutions to this phenomena. 

Nevertheless, there is some information that its missing in this article, as an example the manuscript will benefit of making an uptake comparison among minority ethnic groups in time (end of 2020 when the vaccines where first available until the most recent uptake in May 2021 were are least two/ three covid-19 waves had gone). The reason is because the information was scarce at the beginning and later there was more in the public domain, as well as some communication vaccination programs started late, specially those focused to minority ethnic groups.

Also the manuscript includes the MMAT approach to estimate the risk of bias. It should be provided more information why authors selected this procedure and not others, as an example the GRADE method.

In addition, I'm curious to see if the hesitancy to uptake the covid-19 varies across ages, is this something from the young adults or is something from older adults. Also is unclear what the authors consider a lower covid-19 uptake across the minority ethnic groups, this should be defined in the text. The gap could change from one paper to the other, however some discussion should be available on this sense, what means a lower uptake (e.g. 10%, 15%, 5%). Also, I didn't find in the discussion section the limitations for this article, which is recommended to be included.

Finally, a brief comparison of the UK manuscript results to other Covid-19 uptake differences from  EU markets (and/or US) will be useful to review, just to see if this is really a UK unique issue or this is also happening similarly in other markets, so the issue the authors are addressing is also a global challenge and a global concern which may benefit a broader audience. Lastly, the importance of incrementing the Covid-19 uptake in UK should be mentioned, a discussion on where UK is now in terms of vaccination and what would be any target for UK authorities to achieved with minority ethnic groups is recommended to be included.

Author Response

Many thanks for the helpful feedback provided. We have addressed the comments as follows:

Point 1. There is some information that its missing in this article, as an example the manuscript will benefit of making an uptake comparison among minority ethnic groups in time (end of 2020 when the vaccines where first available until the most recent uptake in May 2021 were are least two/ three covid-19 waves had gone). The reason is because the information was scarce at the beginning and later there was more in the public domain, as well as some communication vaccination programs started late, specially those focused to minority ethnic groups.

Response: A comparison of the most up to date figures have been included.

Point 2. Also the manuscript includes the MMAT approach to estimate the risk of bias. It should be provided more information why authors selected this procedure and not others, as an example the GRADE method.

Response: The rationale for using the MMAT approach has been added to the manuscript

Point 3. In addition, I'm curious to see if the hesitancy to uptake the covid-19 varies across ages, is this something from the young adults or is something from older adults.

Response: The statistics included on row 301 indicate similar patterns of hesitancy in older and younger groups. It was not possible to synthesize findings based on age as several papers did not report this data. We discuss this as a limitation in the discussion section.

Point 4. Also is unclear what the authors consider a lower covid-19 uptake across the minority ethnic groups, this should be defined in the text. The gap could change from one paper to the other, however some discussion should be available on this sense, what means a lower uptake (e.g. 10%, 15%, 5%). Also, I didn't find in the discussion section the limitations for this article, which is recommended to be included.

Response: This has been added to section 4.2 as a limitation

Point 5. Finally, a brief comparison of the UK manuscript results to other Covid-19 uptake differences from  EU markets (and/or US) will be useful to review, just to see if this is really a UK unique issue or this is also happening similarly in other markets, so the issue the authors are addressing is also a global challenge and a global concern which may benefit a broader audience.

Response: This appears to be a global challenge and the discussion section has been updated accordingly.

Point 6. Lastly, the importance of incrementing the Covid-19 uptake in UK should be mentioned, a discussion on where UK is now in terms of vaccination and what would be any target for UK authorities to achieved with minority ethnic groups is recommended to be included.

Response: This has been added to section 4.3 conclusion.